# May the Force be with You: Unified Force-Centric Pre-Training for 3D Molecular Conformations

**Rui Feng**
Computational Science and Engineering
Georgia Institute of Technology
rfeng@gatech.edu

**Qi Zhu**
Computer Science Department
University of Illinois, Urbana-Champaign
qi.zhu.ckc@gmail.com

**Huan Tran**
Materials Science and Engineering
Georgia Institute of Technology
huan.tran@mse.gatech.edu

**Binghong Chen**
Computational Science and Engineering
Georgia Institute of Technology
binghong@gatech.edu

**Aubrey Toland**
Materials Science and Engineering
Georgia Institute of Technology
artoland@gatech.edu

**Rampi Ramprasad**
Materials Science and Engineering
Georgia Institute of Technology
rampi.ramprasad@mse.gatech.edu

**Chao Zhang**
Computational Science and Engineering
Georgia Institute of Technology
chaozhang@gatech.edu

## Abstract

Recent works have shown the promise of learning pre-trained models for 3D molecular representation. However, existing pre-training models focus predominantly on equilibrium data and largely overlook off-equilibrium conformations. It is challenging to extend these methods to off-equilibrium data because their training objective relies on assumptions of conformations being the local energy minima. We address this gap by proposing a force-centric pretraining model for 3D molecular conformations covering both equilibrium and off-equilibrium data. For off-equilibrium data, our model learns directly from their atomic forces. For equilibrium data, we introduce zero-force regularization and forced-based denoising techniques to approximate near-equilibrium forces. We obtain a unified pre-trained model for 3D molecular representation with over 15 million diverse conformations. Experiments show that, with our pre-training objective, we increase forces accuracy by around 3 times compared to the un-pre-trained Equivariant Transformer model. By incorporating regularizations on equilibrium data, we solved the problem of unstable MD simulations in vanilla Equivariant Transformers, achieving state-of-the-art simulation performance with 2.45 times faster inference time than NequIP. As a powerful molecular encoder, our pre-trained model achieves on-par performance with state-of-the-art property prediction tasks.

## 1 Introduction

Representation learning for 3D molecules is a crucial task in the field of AI for science. Recent years have seen significant advancements in Graph Neural Networks [24, 19, 12] and Equivariant Graph Neural Networks architectures [30, 9, 1, 11, 23] to capture interatomic features in 3D molecular

37th Conference on Neural Information Processing Systems (NeurIPS 2023).

conformations, thereby laying foundations for learning powerful representations. Leveraging such architectures, several existing works [33, 10, 17] have explored pre-training on 3D molecular structures on equilibrium data (PCQM4MV2 [16, 15]). Specifically, [33] developed a de-noising approach that learns meaningful information about the equilibrium conformations and their surroundings. [17, 10] leveraged the potential energy of equilibrium conformations to derive a supervised pretraining objective. Such pre-trained models have achieved promising performance on diverse molecular property prediction tasks.

Nonetheless, these existing 3D molecule pre-training models are based on the equilibrium assumption and predominantly overlook off-equilibrium conformations. In this paper, we consider *off-equilibrium* conformations to be those whose atomic forces significantly deviate from zero. In this case, existing work cannot be indiscriminately extended to non-zero forces, because their modeling assumption is fundamentally tied to equilibrium conformations being the local energy minima. The de-noising technique in [33], which is theoretically equivalent to learning forces around equilibrium conformations, may potentially be ill-posed for off-equilibrium conformations. Similarly, the energy-based supervised pretraining objectives in [17, 10] are based on the local minimality of these conformations and cannot be straightforwardly extended to off-equilibrium data. To date, the pre-training of a representation model for 3D molecular conformations that encompasses unifying both equilibrium and off-equilibrium molecular conformations remains largely underexplored. On the other hand, off-equilibrium conformations represent a significant portion of the chemical space, which is crucial for comprehensive modeling of the molecular representation space. For instance, applications such as molecular dynamics (MD) simulations are heavily dependent on the model's ability to accurately represent off-equilibrium data.

In this paper, we incorporate both off-equilibrium and equilibrium conformations into a unified representation learner. We propose a new pre-training model, "**E**quivariant **T**ransformer **O**ptimization and **R**epresentations for **E**quilibrium and **O**ff-equilibrium Molecules" (**ET-OREO**) for 3D molecular conformations. This model integrates both equilibrium and off-equilibrium data from multiple large-scale datasets. Its training objective hinges on *atomic forces*, defined as the negative gradient of a molecule's potential energy with respect to atomic coordinates. Atomic forces exhibit several notable properties:(1) they are *physically well-defined* observable, i.e., the force acting on an atom is determined solely and uniquely from its local environment, defined as the real-space distribution of its neighboring atoms; (2) they are generalizable across various molecules in the sense that atoms from different molecules that have the same local environment should experience the same atomic forces; and (3) they can unify equilibrium and off-equilibrium data, as equilibrium data can be conceptualized as local minima in the latent (configuration) space with zero forces, while off-equilibrium data aid the model in more accurately characterizing the high-energy chemical space beyond equilibrium. Among these points, (2), which is the direct consequence of (1), makes atomic forces fundamentally different from potential energy which is defined for the whole system under consideration (molecules) and can only be determined up to an additional constant. Therefore, a predictive model of atomic forces is transferable, i.e., in principle, it can be used directly for molecules of any size (number of atoms). This advantage makes the "learning atomic forces" fundamentally different from the traditional approach in which a fictitious concept of "atomic energy" must be defined, predicted, and combined to obtain the total potential energy of the whole system [2].

Inspired by this, we develop our force-centric pretraining model that trains on both off-equilibrium and equilibrium molecular conformations. For off-equilibrium data, our model learns directly from their atomic forces, aligning the model gradient with respect to input coordinates with atomic forces. A conceived obstacle in leveraging forces lies in data acquisition: high-accuracy forces require the application of *ab-initio* methods such as Density Functional Theory (DFT) [14, 20]. However, we argue that DFT, despite its time complexity of $O(N^3)$, remains tractable for moderately sized molecules [31]. Additionally, the major computational cost overlap with the potential energy, which is common in existing 3D conformation datasets.

For equilibrium data, we impose zero-force regularizations, reflecting their status as local minima of the potential energy surface. We also introduce random noise into equilibrium conformations and consider the *random noise directions as approximate forces* on the perturbed conformation coordinates. This allows the model to "denoise" the perturbed conformations by gradients. Our approach draws inspiration from the recent success of denoising techniques in molecular learning [33, 13], energy-based supervised pretraining [10, 17], and score-matching generative models [27, 28, 25]. We integrate our gradient-based denoising and zero-force regularization on equilibrium conformations

with off-equilibrium force optimization, thereby providing the model with a unified landscape of molecular structures and the potential energy surface.

For model pre-training, we collated more than 15 million 3D molecular conformations. Our pre-training data leverages three open-domain datasets, including PCQM4Mv2 [16, 15], MD17 [6, 5], ANI1-x [26]. Moreover, we contribute a new dataset, *poly24*, consisting of simulation trajectories of a diverse family of polymers. Pretrained on these diverse sources of equilibrium and off-equilibrium data, our model attained state-of-the-art simulation performance on both MD17 and polymers, achieving efficient and accurate simulations in terms of distributional properties and accuracy relative to DFT calculations. Our model also serves as a potent representation learner for equilibrium data, attaining performance on par with state-of-the-art molecular learning methods that focus exclusively on equilibrium data.

In summary, our contributions are as follows:

- We introduce a novel force-centric molecular conformation pretraining paradigm that trains a unified conformation representation encompassing both equilibrium and off-equilibrium molecules. Our paradigm enables the representation learner to portray not only the equilibrium conformation space but also the extensive off-equilibrium spaces between them.

- Our model achieves highly accurate molecular dynamics (MD) simulations, by efficiently fine-tuning its parameters for use with molecules and polymers. This allows for fast and reliable MD simulation, as our model is able to accurately replicate the *ab initio* forces and distributional properties of conformations. Furthermore, our model has demonstrated comparable performance to state-of-the-art models that are solely focused on equilibrium data in molecular property prediction.

- We provide the community with a diverse set of DFT simulation data comprising a varied set of polymers, which are valuable not only for studying polymer properties such as ring-opening enthalpies but also for the general modeling of molecular forces.

## 2   Related Work

**Machine Learning Forcefield**   In recent years, there has been a surge in the development of deep learning models for molecule forcefields. Two prominent groups of models have emerged: geometric message passing models [24, 19, 12] and the Tensor Field Network (TFN) family [30, 9, 1, 11]. Geometric message-passing models [24, 19, 12] utilize traditional graph neural networks or message-passing networks on pairwise radial features that are translational and rotational invariant. On the other hand, the TFN family models learn SE(3)-equivariant features by leveraging harmonic functions and the irreducible representations of the SE(3) group. NequIP [1] is the state-of-the-art model in this family and has achieved high stability and fidelity on the MD17 dataset.

Along another line, EGNN [23] learns equivariant features by integrating directional distance vectors between atom pairs in their implicit 3-dimensional coordinates. This approach allows for the learning of equivariant features without incurring the computational cost of the TFN family. Our method's backbone model is TorchMDNet [29], which we view as an extension of EGNN. TorchMDNet embeds the implicit 3-dimensional coordinates in a latent high-dimensional space, enhancing the model's capacity to represent 3-dimensional equivariant features. In addition, TorchMDNet leverages the concept of continuous distance filters from SchNet [24] to further increase its representation power.

**Molecular Pretraining**   Inspired by the success of pre-training foundation models in the fields of NLP and CV, there have been several attempts to pretraining for 3D molecular structures [33, 10, 17]. Specifically, the NoisyNode  [33] work adopted the denoising regularization technique of graph neural networks [13] as a pretraining method for equilibrium conformations, achieving state-of-the-art performance on molecular property predictions. [10] and [17] both base their pre-training on the supervised energy data from equilibrium conformations with forces regularization. However, as aforementioned, these existing methods predominantly rely on equilibrium data and cannot be easily extended to off-equilibrium data. While achieving high performance for property prediction tasks, they leave the vast chemical space of off-equilibrium conformations unexplored.

Force-based training over off-equilibrium molecular conformations has been explored in literature. [6, 5, 7, 4, 3] all focus on learning atomic forces and predict energy based on numeric integration, based on the claims that the potential energy and forces have different noise distribution pattern[6, 5] and forces being local and immediate atomic features [4, 3]. Their methods still focus on training on off-equilibrium simulation data from *single molecules*. In supplementary materials, we showed that the joint optimization of potential energy and forces is problematic for multi-molecule settings. We instead propose force-centered and energy-free objectives to learn a unified pretraining model for both off-equilibrium and equilibrium data; and we show that force-centered pre-training improves multi-molecule optimization and generalization.

## 3 Methodology

### 3.1 Problem Setup

Consider a molecule type $x$ that can exist in various 3D conformations. Let $n_x$ represent the number of atoms in molecule $x$. The distribution of the molecule's conformations is represented as $\mathbf{r}_x \in \mathbb{R}^{n_x \times 3}$, and its atom types are represented as $\mathbf{z}_x \in \mathbb{Z}^{n_x}$. The potential energy of a 3D molecule is determined by its conformation $\mathbf{r} \sim \mathbf{r}_x$ and atom types $\mathbf{z}$, denoted as $E = E(\mathbf{r}, \mathbf{z}) \in \mathbb{R}$. The forces applied to the atoms, which are defined as the negative gradient of the potential energy with respect to atomic coordinates, are given by $F = -\frac{\partial E}{\partial \mathbf{r}} \in \mathbb{R}^{n_x \times 3}$. In theory, a stable conformation is one where the potential energy achieves a local minimum with zero forces. In molecular dynamics (MD) simulations, the molecular conformations are moved according to forces and the thermostat of the simulation.

Our goal is to learn an equivariant model $\Phi_\theta(\mathbf{r}, \mathbf{z})$ parameterized by $\theta$ that learns molecular representations from both equilibrium and off-equilibrium conformations. In this paper, $\Phi_\theta(\mathbf{r}, \mathbf{z})$ needs to predict energy-conservative atomic forces. To achieve this, we follow the standard machine learning forcefield paradigm. For simplicity, we omit $\mathbf{z}$ in $E$ and $F$, and we ignore the molecule type index $x$ when referring to general molecules. We use $\Phi : (\mathbb{R}^{n \times 3}) \times \mathbb{Z} \to \mathbb{R}$ to output the potential energy and take $-\nabla\Phi_\theta$ as forces. To distinguish between equilibrium and off-equilibrium conformations, we define the equilibrium set of conformations as $\mathscr{E} := \{\mathbf{r} : F(\mathbf{r}) = 0\}$ and the off-equilibrium set $\mathscr{S}$ as the complementary set where the forces are non-zero.

### 3.2 Joint Training of Forces on Equilibrium and Off-equilibrium Data

Our methodology is centered on forces optimization. For off-equilibrium conformations $x \sim \mathscr{S}$ with non-zero forces, we directly optimize on the atomic forces by minimizing $\|-\nabla\Phi_\theta(\mathbf{r}_x) - F(\mathbf{r}_x)\|_2^2$. For equilibrium molecular conformations $x \sim \mathscr{E}$, we assume their forces to be zero and impose a zero-force regularization: $\|-\nabla\Phi_\theta(\mathbf{r}_x)\|_2^2$. However, this objective gives the model little knowledge of the conformation structures in the neighborhood of $\mathbf{r}_x$. To better inform the model of the forcefield around equilibrium conformations, we further use a *de-noising equilibrium regularization*, where we add Gaussian noise on equilibrium conformations and use the noise direction as noisy forces:

$$\mathbb{E}_{\varepsilon \sim \mathcal{N}(0,\sigma^2)} \left[ \|\nabla_{\mathbf{r}_x}\Phi(\mathbf{r}_x - \varepsilon) - \varepsilon\|_2^2 \right], \qquad x \sim \mathscr{E}. \tag{1}$$

The rationale of the above force-guided denoising objective is as follows. The perturbed conformation, denoted as $\mathbf{r}_x - \varepsilon$, could either approximate a high-energy, off-equilibrium confirmation or, alternatively, represent an unphysical state. In the former scenario, the conformation is anticipated to relax back to the proximate local minimum, $\mathbf{r}_x$, thereby yielding forces that are consistent with the direction $\varepsilon$. Conversely, in the latter situation, the learning model is still sufficiently equipped to maintain robust representations, thereby enabling a return to a stable conformation from any unphysical deviations. More formally, (1) can be interpreted as learning approximate forcefield around equilibrium conformations, as will be discussed in Section 3.3.

Combining forces optimization on off-equilibrium data and zero-force regularization and de-noising objective on equilibrium data, we have the unified force-centric pre-training objective

$$\mathbb{E}_{x \sim \mathscr{E}} \left[ \underbrace{\|\nabla_{\mathbf{r}_x}\Phi(\mathbf{r}_x)\|_2^2}_{\text{zero-force regularization}} + \underbrace{\mathbb{E}_{\varepsilon \sim \mathcal{N}(0,\sigma^2)} \left[ \|\nabla_{\mathbf{r}_x}\Phi(\mathbf{r}_x - \varepsilon) - \varepsilon\|_2^2 \right]}_{\text{de-noising equilibrium}} \right] + \mathbb{E}_{x \in \mathscr{S}} \left[ \underbrace{\|F(\mathbf{r}_x) - \nabla_{\mathbf{r}_x}\Phi(\mathbf{r}_x)\|_2^2}_{\text{forces optimization}} \right],$$

$$\tag{2}$$

where the first expectation over $x \sim \mathscr{E}$ samples equilibrium conformations and imposes a zero-force regularization and the denoising objectives on the atomic coordinates. The expectation over $x \sim \mathscr{S}$ samples off-equilibrium conformations and optimizes the model gradient with forces.

## 3.3 On the De-noising Equilibrium Objective

Objective function (1) can be viewed as learning an approximate forcefield around equilibrium data. In fact, by well-known results in de-noising score-matching [32, 27, 28], the objective

$$\mathbb{E}_{\varepsilon \sim \mathcal{N}(0,\sigma^2)} \left[ \left\| \nabla_{\mathbf{r}} \Phi(\mathbf{r} - \varepsilon) - \varepsilon/\sigma^2 \right\|_2^2 \right] \tag{3}$$

is equivalent to the score-matching-like objective

$$\mathbb{E}_{q_\sigma(\tilde{\mathbf{r}})} \left[ \left\| \nabla_{\tilde{\mathbf{r}}} \Phi(\tilde{\mathbf{r}}) - \nabla_{\tilde{\mathbf{r}}} \log q_\sigma(\tilde{\mathbf{r}}) \right\|_2^2 \right], \quad q_\sigma(\tilde{\mathbf{r}}|\mathbf{r}) := \mathbf{r} + \varepsilon, \tag{4}$$

which by [33] is equivalent to learning implicit forces around $\mathbf{r}$, assuming $\mathbf{r}$ is an local minimizer of the energy. Hence, (1) is equivalent to learning forces around equilibrium conformations up to a scaling constant. This proof, however, cannot be naïvely extended to molecules whose forces are non-zero, which is the driving motivation for us to supplement the de-noising objective with forces from off-equilibrium conformations.

While [33] used the same argument for their de-noising objective, our implementation is different. They used a prediction head for predicting the noises on top of an Equivariant Transformer, while we directly predicted the noise with forces predicted by the model gradient. Our design has several unique advantages: 1) Forces principally govern atomic movements. By formulating the de-noising objective with predicted forces, our model has a physical interpretation where perturbed conformations, be it high-energy off-equilibrium conformations or unphysical conformations, could relax back to stable and physical conformations. This property is especially helpful in MD simulations, as will be shown in Section 4. 2) We can unify the prediction of the forces for both equilibrium and off-equilibrium data, providing the model with a consistent energy landscape across conformations of different molecules and states.

## 3.4 Model Pre-Training and Fine-Tuning

**Model Architecture.** We use TorchMDNet [29] (also known as Equivariant Transformer) as our molecule encoder $\Phi_\theta$, following [33]. TorchMDNet is one of the best-performing models in terms of predicting molecular properties and atomic forcefields. While [1] achieves higher accuracy in molecular prediction tasks, we find TorchMDNet more favorable for pre-training due to its expressivity and better computational efficiency.

**Pre-training.** Our model has 8 layers and 256 embedding sizes, 8 attention heads, and a radial basis function (RBF) dimension of 64, consistent with [29] and [33]'s best-performing model. The model parameters are optimized with the AdamW[22, 18] optimizer with a peak learning rate set as $1e-4$. The learning rate is scheduled with 10,000 warmup steps that gradually increase the learning rate from $1e-7$ to $1e-4$, and afterward will decrease with a multiplier of 0.8 after 30 validation epochs of non-decreasing validation loss. On every 500 training steps, one validation epoch is performed. The model is trained with a batch size of 32 samples for 3 epochs for 468,750 gradient steps. For the denoising objective, the variance of noise $\sigma^2$ is set to be 0.04.

**Model Fine-tuning.** The pre-training stage provides the model with a good initial representation of molecules in general but is not necessarily optimal for each specific dataset or task. For our experiments, the model is further fine-tuned on the target dataset to optimize task-specific performance. For ET-OREO, the training/testing split of overlapping datasets in both the pre-training and fine-tuning stages are consistent so that during fine-tuning, the test set will not include any data seen during training.

## 3.5 Pre-Training Data

Our paper focuses primarily on the structures of organic molecules and polymers in a vacuum. This approach enables us to maintain consistent learning of quantum mechanical interactions between

atoms within a uniform environment. For this purpose, we use three public 3D molecular conformation datasets MD17, ANI1-x, and PCQM4Mv2; and we also create a new polymer simulation dataset, detailed as follows.

**Poly24: MD Simulations for Polymers.** We contribute *poly24*, a DFT-based MD simulation dataset for polymers. [1] We generated DFT simulation data for 24 types of polymer families, broadly categorized into cycloalkanes, lactones, ethers, and others. Each polymer family consists of a cyclic monomer and its ring-opening polymerizations. Within ring-opening polymerization, a ring of the cyclic monomers is broken and then the "opened" monomer is added to a long chain, forming a polymer chain. The details on data generation can be found in the Appendix. In total, we run generally 10 DFT simulations for different initialization of each $L$-loop ($L = 1, 3, 4, 5, 6$) polymer across the 24 types of polymers. We have 1311 DFT trajectories and 6,552,624 molecular conformations. Only polymers with less than or equal to 64 atoms are used for pre-training, totaling 3,851,540 conformations. The remaining larger-polymer conformations are used for fine-tuning and testing.

**MD17, ANI1-x, and PCQM4Mv2.** In addition to our own *poly24*, we have utilized three existing public datasets, namely MD17[6, 5], ANI1-x[26], and PCQM4Mv2[16, 15] for our model pre-training.

| Dataset | # Conformations | Equilibrium | Off-equilibrium |
|---------|-----------------|-------------|-----------------|
| PCQM4Mv2 | 3,378,606 | ✓ | ✗ |
| ANI1x | 4,956,005 | ✓ | ✓ |
| MD17 | 3,611,115 | ✗ | ✓ |
| poly24 | 3,851,540 | ✗ | ✓ |
| Total | 15,718,279 | ✓ | ✓ |

Table 1: Datasets used in our model pre-training process.

These datasets contain small organic molecules in a vacuum, and property prediction for such molecules is an area of great interest to the cheminformatics community. The machine learning for molecules community has also extensively studied and benchmarked these datasets. Table 1 summarizes all the dataset used in the pre-training stage. In total, we have more than 15 million samples from MD17, ANI1-x, PCQM4Mv2, and Poly24, covering equilibrium and off-equilibrium conformations for diverse organic molecules.

## 4  Experiments

### 4.1  MD Simulations for Small Molecules on MD17

**Setup** After pre-training our model ET-OREO, we further finetune it on the MD17 dataset to validate the performance of molecular simulations using the forces predicted by our fine-tuned model. We run simulations with our model's predicted forces with a Nose-Hoover thermostat, with 500K temperature and 0.5fs timestep for 600k steps. The simulation setting is the same as [8], which benchmarked popular deep learning forcefields for MD simulations. We use the ASE package for the implementation of molecular simulation environment [21] with a characteristic time for the Nose-Hoover thermostat set to 25fs.

**Baselines and Metrics.** Following [8], we report 3 metrics:

- *Forces MAE*, which is the mean absolute error of forces on the DFT trajectories;

- *Stability*, which measures how long the simulation can run before blowing up. According to [8], this is detected when the radial distance function (RDF) deviates from the reference simulation by a threshold (0.10Å).

- *h(r) MAE*, with $h(r)$ representing the distribution of interatomic distances during simulation. According to [8], MAE here is calculated as the $l1$-norm between the reference distribution and the predicted distribution.

We compare ET-OREO with the best-performing models reported by [8]. Furthermore, ET-OREO is compared with TorchMDNet [29] trained on MD17 from scratch, and ET-ORE, which is an ablation version of ET-OREO without the zero-force and de-noising regularizations. Consistent with previous

---

[1]Dataset will be made available upon publication.

| Molecule | Metric | DimeNet | GemNet-T | GemNet-dT | NequIP | TorchMDNet | ET-ORO | ET-OREO |
|---|---|---|---|---|---|---|---|---|
| Aspirin | Force ($\downarrow$) | 10.0 | 3.3 | 5.1 | 2.3 | 7.4 | 4.2 | **1.0** |
| | Stability ($\uparrow$) | $54_{(12)}$ | $72_{(50)}$ | $192_{(132)}$ | $300_{(0)}$ | $102_{(45)}$ | $94_{(42)}$ | $300_{(0)}$ |
| | $h(r)$ ($\downarrow$) | $0.04_{(0.00)}$ | $0.04_{(0.02)}$ | $0.04_{(0.01)}$ | $0.02_{(0.00)}$ | $0.04_{(0.00)}$ | $0.04_{(0.00)}$ | $0.02_{(0.00)}$ |
| Ethanol | Force | 4.2 | 2.1 | 1.7 | 1.3 | 5.6 | 3.1 | **1.0** |
| | Stability | $26_{(10)}$ | $169_{(98)}$ | $300_{(0)}$ | $300_{(0)}$ | $121_{(34)}$ | $300_{(0)}$ | $300_{(0)}$ |
| | $h(r)$ | $0.15_{(0.03)}$ | $0.10_{(0.02)}$ | $0.09_{(0.00)}$ | $0.08_{(0.00)}$ | $0.12_{(0.01)}$ | $0.10_{(0.00)}$ | $0.03_{(0.00)}$ |
| Naphthalene | Force | 5.7 | 1.5 | 1.9 | 1.1 | 3.3 | 2.0 | **0.9** |
| | Stability | $85_{(68)}$ | $8_{(2)}$ | $25_{(10)}$ | $300_{(0)}$ | $50_{(20)}$ | $25_{(9)}$ | $300_{(0)}$ |
| | $h(r)$ | $0.10_{(0.01)}$ | $0.13_{(0.00)}$ | $0.12_{(0.01)}$ | $0.12_{(0.01)}$ | $0.12_{(0.00)}$ | $0.11_{(0.00)}$ | $0.03_{(0.00)}$ |
| Salicylic Acid | Force | 9.6 | 4.0 | 4.0 | 1.6 | 4.7 | 2.5 | **0.9** |
| | Stability | $73_{(82)}$ | $26_{(24)}$ | $94_{(109)}$ | $300_{(0)}$ | $60_{(69)}$ | $94_{(58)}$ | $300_{(0)}$ |
| | $h(r)$ | $0.06_{(0.02)}$ | $0.08_{(0.04)}$ | $0.07_{(0.03)}$ | $0.03_{(0.00)}$ | $0.06_{(0.02)}$ | $0.05_{(0.01)}$ | $0.02_{(0.00)}$ |

Table 2: Simulation results on MD17. For all results, force MAE is reported in the unit of [meV/Å], and stability is reported in the unit of [ps]. The distribution of interatomic distances $h(r)$ MAE is unitless. FPS stands for frames per second. For all metrics ($\downarrow$) indicates the lower the better, and ($\uparrow$) indicates the higher the better. The first group of methods is taken from [8]. The second group of methods is our new baselines, including TorchMDNet [8], ET-ORO, and ET-OREO. These models share the same architecture and have the same FPS.

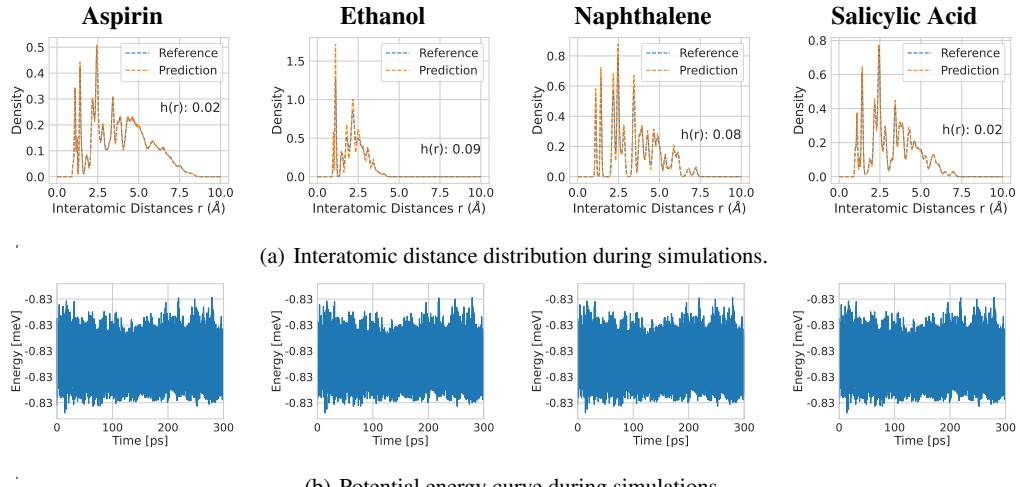

(a) Interatomic distance distribution during simulations.

(b) Potential energy curve during simulations.

Figure 1: Illustration of simulation statistics for ET-OREO on MD17 dataset. Figure (a): distributions of interatomic distances during reference simulation and simulation with ET-OREO predictions finetuned on MD17. Figure (b): the curve of potential energy predicted by ET-OREO during simulation.

simulation benchmark [8], TorchMDNet, ET-ORE, and ET-OREO are fine-tuned with 9,500 conformations for training and 500 for validation. The training and validation conformations are sampled from the same training data used in the pre-training of ET-OREO, and the metrics are reported on the rest of MD17 data unseen to both the fine-tuning and the pre-training of ET-OREO and ET-ORE. Furthermore, in our implementation, we focus solely on forces optimization for TorchMDNet. We show in Appendix that this improves forces accuracy without sacrificing the model's ability to predict the potential energy.

**Results.** In Table 2, we compare the performance of ET-OREO with baseline models. We make the following key observations: (1) On all of the simulations, we have achieved a lower interatomic distance distribution $h(r)$ than the previous best-performance model NequIP reported by [8]. Particularly, we reduced $h(r)$ for aspirin and salicylic acid by half, 10% for ethanol, and 18% for naphthalene. While NequIP [1] can achieve stable and accurate MD simulations by training from scratch on MD17, it suffers from a lower FPS as it is computationally expensive. In contrast, ET-OREO can achieve both fast and high-quality MD simulations. (2) ET-OREO improves forces accuracy over TorchMDNet by over three times, achieving state-of-the-art MAE on all four tested molecules. The major difference between ET-OREO and TorchMDNet is that ET-OREO is pre-trained with our force-centric objective before fine-tuning on MD17 data. Such significant performance gains show the benefits of pre-training over diverse equilibrium and off-equilibrium conformations. (3) ET-OREO produces

stable simulations and accurate interatomic distance distributions. ET-OREO is able to stably run simulations on all four molecules with accurate interatomic distance distribution $h(r)$ compared to the reference trajectory in MD17. Figure 1 visualizes the close approximation of ET-OREO predicted $h(r)$ compared with reference data, and that ET-OREO can produce energy-conserving simulations that sample around the equilibrium.

**Regularization on Equilibrium Conformations is Vital for Simulations.** We found that training TorchMDNet itself on MD17 cannot produce stable MD simulations. Meanwhile, the performance of ET-ORE shows that pre-training with forces only helped with higher forces accuracy. However, ET-ORE still does not perform well for MD17 simulations. Except for ethanol, ET-ORE cannot produce stable MD simulations, despite consistently improved forces accuracy. This is in line with [8]'s observation that higher forces accuracy does not always guarantee simulation performance. The difference between ET-ORE and ET-OREO is that the latter incorporates stable zero-force conformations and a de-noising regularization objective. The additional regularization on the conformation space proves vital to stable and accurate simulations.

**ET-OREO achieves accurate MD simulation with 3 times inference speed.** From Table 2, NequIP reported similar simulation performance to ET-OREO without need for pre-training. However, NequIP has a significantly slower inference time. The NequIP model in Table 2 has an inference time of approximately 119ms per step on NVIDIA V100 with 1.05 million model parameters. [8] In comparison, ET-OREO has an inference time of 48.5ms on the same hardware with 6.87 million parameters. Hence, we have achieved state-of-the-art simulation performance with a 2.45 faster inference speed.

## 4.2 Simulation on Large-loop Polymers

The vast majority of our model's training data is small molecules. In this section, we investigate the model's ability to generalize to larger molecules, consisting of 15-loop polymers unseen in the training data.

**Setup** We finetune our model on the forces of small polymers (5-loop or less) in the poly24 dataset for one epoch with a learning rate of 0.0001. For testing the model's MD simulation performance on large polymers, we take the larger 15-loop polymers from the poly24 dataset and run MD simulations on them with ET-OREO-fine-tuned forcefields for a maximum of 600K steps. The number of atoms for these 15-loop polymers are, respectively: 360, 360, 180, and 240. In the training data, we have at most molecules consisting of $\approx 100$ atoms. Therefore, ET-OREO is required to perform accurate MD simulations on unseen polymers, testing both its simulation and generalization ability.

| ID | Polymer Name | # atoms | SMILES | Monomer | Polymer |
|----|--------------|---------|--------|---------|---------|
| CK | cyclooctane | 360 | [*]CCCCCCCC[*] | | |
| OTH | n-alkane substituted $\delta$-valerolactone | 360 | [*]OC(CCC)CCCC([*])=O | | |
| LAC1 | $\gamma$-butyrolactone | 180 | [*]OCCOCC([*])=O | | |
| LAC2 | 3-methyl-1,4-dioxan-2-one | 240 | [*]OCCOC(C)C([*])=O | | |

Table 3: Details of 15-loop polymers we used for simulations, including their ID, polymer name, number of atoms for the 15-loop polymer, SMILES string, and rdkit visualizations.

**Results.** Figure 2 (a) visualize the comparison results between ET-OREO predicted forces and DFT reference data. For all 15 loop polymers, ET-OREO obtains highly accurate forces to DFT calculations, with 0.01 eV/ MAE in energy and close to zero cosine distance. Hence, ET-OREO can perform MD simulations on multiple types of polymers with uniformly high correlation with DFT references. The test 15-loop polymers contain up to 360 atoms, unseen in the training data. This shows that ET-OREO can extrapolate well to large unseen polymers with known monomers. Furthermore, in practice, ET-OREO only needs training data from small polymers, which are cheap to generate *ab initio* data and thus greatly reduce the cost of DFT simulation.

Figure 2 (b) shows the potential energy curves during simulation. The potential energy curves indicate that the polymer in the simulation converges to a stable near-equilibrium distribution quickly after

initial fluctuations. Furthermore, ET-OREO can explore different possible conformation states of the polymers, as illustrated in Figure 2 (c).

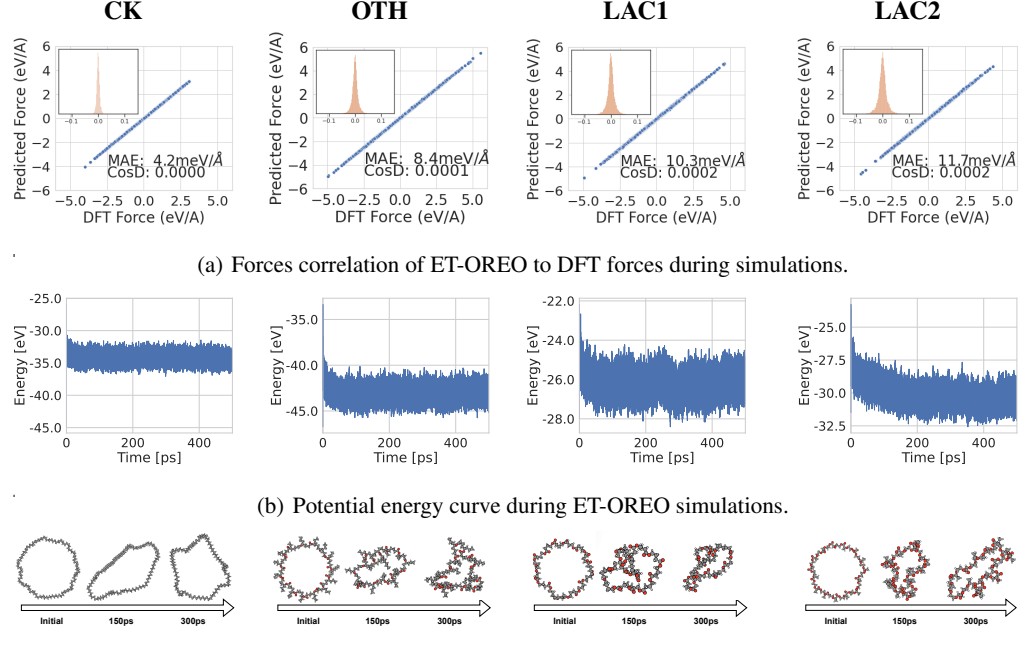

(a) Forces correlation of ET-OREO to DFT forces during simulations.

(b) Potential energy curve during ET-OREO simulations.

(c) Sampling of conformation spaces during ET-OREO simulations.

Figure 2: Accuracy and Robustness of ET-OREO in MD simulations for large polymers. In Figure (a) it is observed that ET-OREO forces are highly correlated with DFT forces during its own MD simulations. Figure (b) shows the RMSD and potential energy curves of ET-OREO during MD simulations. The curves suggest that ET-OREO simulations experience equilibration. Figure (c) shows the visualization of sample conformations captured at various time points during the simulation.

## 4.3 Property Prediction on QM9

To test ET-OREO's ability to encode equilibrium conformation for property prediction tasks, we follow the same experiment setting as NoisyNode [33] on QM9. [33] exclusively pre-trained on equilibrium conformations from PCQM4Mv2 with a de-noising objective. In comparison, ET-OREO has access to

|  | TorchMDNet | NoisyNode | ET-OREO |
|---|---|---|---|
| $\varepsilon_{\text{HOMO}}$ | 20.3 | **15.6** | 16.8 |
| $\varepsilon_{\text{LUMO}}$ | 17.5 | **13.2** | 14.5 |
| $\Delta\varepsilon$ | 36.1 | **24.5** | 26.4 |

Table 4: Fine-tuning on HOMO-LUMO properties on QM9. Metrics are MAE in meV.

larger off-equilibrium data supervised by forces. We follow the exact same fine-tuning setup as [33] save the de-noising regularization in the fine-tuning stage. In Table 4, we compare ET-OREO with NoisyNode and TorchMDNet [29] on HOMO-LUMO properties prediction on QM9. We report the performance of NoisyNode trained on the TorchMDNet encoder reported in [33], hence all three models have the same encoders, except that TorchMDNet trains from scratch, and NoisyNode and ET-OREO fine-tune from their pre-trained parameters. ET-OREO improves the performance of property prediction by ∼30% compared to TorchMDNet trained from scratch, implying that our pre-training paradigm provided the model with useful information about the quantum mechanical properties of equilibrium conformations. Our performance is on par with NoisyNode, and unlike in simulation experiments, we observe no accuracy improvement by the inclusion of the rich off-equilibrium conformation. We speculate that such equilibrium property prediction tasks benefit little from additional information on off-equilibrium conformation. We want to particularly point out that, despite the pre-training effort, NoisyNode and ET-OREO both require extensive fine-tuning with a large amount of data (more than 10,000) and training epochs (roughly 150 epochs for convergence) for optimal performance. An important future improvement in molecular pre-training should be in improving the data and time efficiency of fine-tuning.

# 5 Discussion

We presented ET-OREO, a pre-trained model for 3D molecules on $\approx$15M 3D conformations of both equilibrium and off-equilibrium states. We unified the pre-training of molecular conformations of different sources and states with a force-centric pre-training objective. With our pre-training model, we achieved state-of-the-art performance on MD simulations, with both high force accuracy and simulation efficiency. As a potent encoder for conformations, our model also attained a state-of-the-art level of performance on property prediction on equilibrium data.

Our current model is limited mostly to single and small molecules in a vacuum, leaving more complex molecular environments as future work. We also deem it a promising direction to explore more model architectures and larger model sizes. Furthermore, our de-noising objective on equilibrium data does not leverage the model's learned information on atomic forces. It will be interesting to study the possibility of a more sophisticated coupling of equilibrium and off-equilibrium optimization, to make the model progressively leverage its quantum mechanical knowledge for equilibrium data.

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
