# A  Poly24 Dataset

Poly24 is a dataset provided by Density Function Theory and is designed to calculate the enthalpy change in ring-opening polymerization ($\Delta H^{\text{ROP}}$), a critical thermodynamic principle dictating ring-opening polymerization processes. This process involves breaking a cyclic monomer ring and attaching the "opened" monomer to an extensive chain, ultimately forming a polymer chain. The dataset was generated using Molecular Dynamics (MD) simulations of various monomer and polymer models at a consistent level of DFT [6, 9] computations.

Leveraging the Polymer Structure Predictor (PSP) package [13], various polymer models were generated from a given cyclic monomer. Each model was obtained by multiplying the monomer with a small integer $L$, for instance, $L = 3, 4, 5$, and 6, thereby creating a loop of size $L$ (with larger loops more accurately modeling polymers). For every monomer or polymer model, approximately ten or more maximally diversified configurations were selected as the starting points for the MD simulations based on Density Functional Theory (DFT).

Given that all of our models are non-periodic, we utilized the $\Gamma$-point version of the Vienna Ab initio Simulation Package (VASP) [10, 11], applying a plane wave basis set with kinetic energy reaching up to 400 eV for the depiction of Kohn-Sham orbitals. The interactions between ions and electrons were calculated using the Projector Augmented Wave (PAW) method [2], while the exchange-correlation (XC) energies were determined utilizing the Generalized Gradient Approximation (GGA) in the form of the Perdew-Burke-Ernzerhof (PBE) functional [12].

Table 1 presents the polymers utilized in our study, which consists of 24 distinct polymer types. These types are broadly categorized into cycloalkanes, ethers, lactones, and others, in alignment with the classification system applied in [16]. On average, the dataset encompasses 10 DFT trajectories for both monomers and polymers. The polymers are produced by polymerizing the monomers, thereby yielding a thorough dataset for each type of polymer.

# B  Multi-molecule Forces Training

In this section, we provide an empirical justification for a force-centric framework for training a unified machine learning model for molecules. In recent studies in the machine learning forcefield community [1, 15, 14, 4, 8, 5], the paradigm is often learning both the potential energy and forces, where the forces are taken as the derivative of the potential energy. However, here we note an inherent inconsistency of this approach in a greedy optimization framework caused by drastically different distributions of potential energy and forces when multiple molecules are considered in the dataset.

[3] noticed a different noise signature for energy and forces, arguing that learning on forces alone is better than learning on potential energy. In recent studies in the machine learning community [1, 15, 14, 4, 8, 5], the paradigm is often learning both the potential energy and forces, where the forces are taken as the derivative of the potential energy. However, here we note an inherent consistency of this approach in a greedy optimization framework caused by drastically different distributions of potential

| Index | Type | # Atoms | Monomer | Polymer |
|-------|------|---------|---------|---------|
| **CK1** | cyclopropane | 9 | | |
| **CK2** | cyclobutane | 12 | | |
| **CK3** | cyclopentane | 15 | | |
| **CK4** | cyclohexane | 18 | | |
| **CK5** | cycloheptane | 21 | | |
| **CK6** | cycloctane | 24 | | |
| **ETH** | ethylene oxide | 7 | | |
| **LAC1** | $\gamma$-butyrolactone | 9 | | |
| **LAC2** | $\gamma$-butyrolactone | 12 | | |
| **LAC3** | 1,4-dioxan-2-one | 13 | | |
| **LAC4** | $\delta$-valerolactone | 15 | | |
| **LAC5** | 3-methyl-1,4-dioxan-2-one | 16 | | |
| **LAC6** | $\beta$-methyl-$\delta$-valerolactone | 18 | | |
| **LAC7** | $\delta$-caprolactone | 18 | | |
| **LAC8** | $\delta$-decalactone | 30 | | |
| **LAC9** | (-)-Menthide | 30 | | |
| **OTH1** | n-alkane sub $\delta$-valerolactone | 18 | | |
| **OTH2** | $\alpha$-Methylene-$\gamma$-butyrolactone | 13 | | |
| **OTH3** | n-alkane sub $\delta$-valerolactone | 21 | | |
| **OTH4** | n-alkane sub $\delta$-valerolactone | 24 | | |
| **OTH5** | n-butyl $\delta$-valerolactone | 27 | | |
| **OTH6** | n-alkane sub $\delta$-valerolactone | 30 | | |
| **OTH7** | n-alkane sub $\delta$-valerolactone | 33 | | |
| **OTH8** | n-alkane sub $\delta$-valerolactone | 42 | | |

Table 1: Polymers in our used poly24 dataset. We follow [7] and classify polymers into 4 broad categories: cycloalkanes, lactones, ethers, and others.

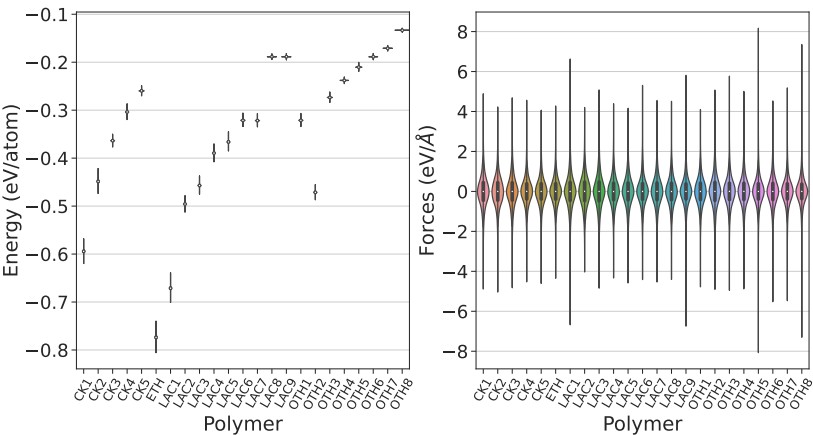

Figure 1: The distribution of per-atom potential energies and forces across different polymers. Forces exhibit similar distributions for various types of polymers, whereas the per-atom potential energy does not.

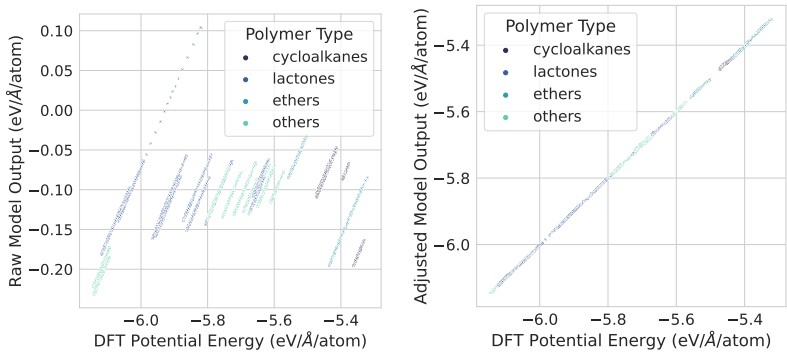

(a) Model Output vs DFT Potential Energy.

(b) Adjusted Model Output vs DFT Potential Energy.

Figure 2: Energy Prediction Performance. Figure 2(a): model output plotted against DFT potential energy in eV/Å/atom. Figure 2(b): model predicts accurate potential energy after applying linear transforms for each polymer.

|  | eparate | Energy + Forces | Normed Energy + Forces | Forces | Pre-trained Forces |
|---|---|---|---|---|---|
| Relative MAE | 1 | 5.62 | 2.31 | 0.76 | 0.53 |
| Absolute MAE (kcal/mol) | 0.13 | 0.73 | 0.30 | 0.10 | 0.07 |

Table 2: Comparison of multi-molecule training strategies with TorchMDNet on MD17. Results are reported in relative MAE in comparison with separate TorchMD-Net models on different molecules.

energy and forces when multiple molecules are considered in the dataset.

In Table 2, we compare the performance of different strategies for multi-molecule training with separately trained models. The tests use the same backbone model, Torch-MDNet [15]. The different strategies include

- Energy + Forces: joint optimization on energy (divided by the number of atoms) and forces,

- Normed Energy + Forces: joint optimization on normalized energy and forces,

- Forces Only: optimization on forces only, and

- Pre-trained Forces: optimization of forces only, while the model is initialized from a pre-trained forcefield model on poly24.

In Table 2 we show the relative performances of different strategies for joint training on all molecules from MD17. The Relative MAE in Table 2 is calculated as $(e - e_{\text{sep}})/e_{\text{sep}}$, where $e$ is the mean absolute error and $e_{\text{sep}}$ is the baseline mean absolute error evaluated with TorchMDNet trained on separate molecules [15]. Jointly training on multiple molecules with both energy and forces would result in a deterioration of performance of 5.62 times. Even if the energies are normalized per molecule, which means that energies are supposed to have same means and variances across different molecules, jointly training on energies and forces still result in 2.31 times deterioration on average, suggesting more inherent conflicts in the joint optimization of energy and forces for multiple molecules. In comparison, by simply focusing exclusively on forces, the performance can be improved to 0.76 of the separately trained models. With the model pre-trained on polymers and further finetuned on MD17, the performance can further achieve an MAE of 0.07 kcal/mol, which is about half of the error of the separately trained TorchMDNet and achieves the state-of-the-art accuracy on MD17.

Our experiments provide empirical support that forces are a more essential and informative feature for different types of molecules. Forces

## C   Energy Prediction

The above section provides a justification of force-centric optimization. However, the potential energy is an important value indicating the energy state of the molecule. In this section, we further show that training exclusively on forces do not lose us much information on the energy, as the energy is essentially the integral of the forces.

In Figure 2(a), we plot the model output of TorchMDNet trained exclusively on forces on all polymers in poly24 against ground-truth potential energies. Although the model is only trained on forces, the model output is naturally linearly correlated with ground-truth potential energy. For each polymer, if we take 4 ground-truth conformations and potential energies and estimate their mean and variance then re-scale the model output accordingly, we obtain Figure 2(b), showing perfect linear correlation and high accuracy on potential energy for all polymers. This shows that the model trained only on forces can be adjusted to predict potential energies with estimation from only a few samples.

# D    Simulation Evaluation