# OpenReview forum: "May the Force be with You: Unified Force-Centric Pre-Training for 3D Molecular Conformations"
_NeurIPS.cc/2023/Conference — NeurIPS 2023 poster_

### Official Review · Reviewer_1y4N · 2023-07-06

**Soundness:** 3 good
**Presentation:** 3 good
**Contribution:** 3 good
**Rating:** 7
**Confidence:** 4

**Summary:**

1. proposing a pretraining framework for conformation space including equilibrium and off-equilibrium.
2. this method also shows highly accurate in MD or relax, including molecules and polymers.
3. providing the community with a diverse set of DFT dataset of polymers.

**Strengths:**

1. This paper is well-written and well-organized, effectively presenting the significance of force-centric pretraining for molecular systems. The proposed framework demonstrates comprehensive and impressive results in molecular dynamics simulations of small molecules and polymers, as well as in quantum chemistry property predictions.
2. The authors also introduce an innovative concept in their work: the de-noising equilibrium balance and off-equilibrium co-training method.

**Weaknesses:**

1. As previously mentioned, I am particularly intrigued by the efficiency of the de-noising equilibrium tasks. It would be better to conduct an detail ablation study of de-noise equillibrium and force regularization.


**Questions:**

1. Would it be possible to incorporate the PCQM4Mv2 homo-lumo prediction results in the downstream analysis? This benchmark is well-defined and provides a competitive comparison for evaluating the performance of various models.

2. About code and data access, I believe that the Poly24 dataset is of significant value to the research community for future investigations.

**Limitations:**

In the discussion section, it is mentioned that both the backbone and time efficiency play crucial roles in the performance of the proposed AI model.
Besides I recommend incorporating partial charge prediction during the pretraining or downstream stages of your framework, if feasible. This addition could potentially enhance the overall effectiveness and applicability of your model.

---

> ### Author Rebuttal · Authors · 2023-08-10
>
> Thank you for your valuable insight! In our separate work, we have indeed found that including partial charge in pre-training where available helps property prediction tasks. We are actively investigating this and more strategies to make the pre-training more efficient.
>
> In the rebuttal period, we are unable to finish complete experiments on pre-training and fine-tuning without the de-noising objective. We agree that it would be valuable to investigate the ablated effects of each objective, and we will do so to make the paper more technically solid.

---

### Official Review · Reviewer_8Ywg · 2023-07-07

**Soundness:** 3 good
**Presentation:** 3 good
**Contribution:** 2 fair
**Rating:** 4
**Confidence:** 4

**Summary:**

This paper incorporates off-equilibrium molecular conformations into consideration, in which the energy is not minimized. This paper defines pre-training objectives for force field learning on sampled off-equilibrium molecular conformations. Experimental results validate the effectiveness of pre-training on additional off-equilibrium data.

**Strengths:**

(1) The presentation of the paper is quite clear and easy to follow. The writing is well-organized;

(2) The proposed idea is quite interesting and necessary. How to utilize off-equilibrium data is a useful and important research question for pre-training 3D molecular property prediction. This reviewer agrees with the significance of the proposed idea.

**Weaknesses:**

(1) The novelty of the proposed method is not very high. Considering there are already published papers researching denoised pretraining strategies on 3d molecules, which lowers the novelty of the proposed method a bit. Then the major contribution of the proposed method is mainly about introducing additional data. This reviewer suggests that this paper might be more suitable for a domain journal;

(2) The contribution of this paper might not be enough. Currently, the major contribution includes introducing additional data, and additional pre-training objectives. There are some important works like SCN (Spherical channel networks for modeling atomic interactions) utilizing separate force predictions. So separately predicting forces cannot be regarded as an important contribution;

(3) The experimental results do not demonstrate the effectiveness of the proposed method. It seems that the proposed method seldom outperforms previous baseline methods, which makes this reviewer feel a little bit confused.

**Questions:**

It seems the reported results on MD17 are not that superior. Probably this reviewer misunderstands the experimental tables but it seems the sota method is usually not the proposed method. Does the proposed method outperform the previous baseline methods?

**Limitations:**

The major limitation of the proposed method is that the force prediction might only play a role as a regularizer since learning force field is also a very challenging task since force is a vector feature instead of a scalar feature like energy.

---

> ### Author Response · Authors · 2023-08-17
>
> We would like to kindly remind the reviewer to read our response that we have made available for all reviewers in response to questions on the presentation of the paper. Thank you!

---

> ### Author Response · Authors · 2023-08-20
>
> We would like to remind the reviewer to please take a look at our previous reply and comments, especially regarding the contribution of the paper and further experiments and explanations of the improved performance of ET-OREO.

---

### Official Review · Reviewer_EQHq · 2023-07-08

**Soundness:** 2 fair
**Presentation:** 2 fair
**Contribution:** 3 good
**Rating:** 5
**Confidence:** 5

**Summary:**

The paper propose a set of additional objectives for training machine learning interatomic potentials, in particular for equilibrum conformations as well as a denoising term. They show that this improves stability, when pretrained on a large dataset. They also introduce a new dataset of polymers.

**Strengths:**

The improvement from ET-ORE to ET-OREO is interesting and surprising. It doesn't really make sense that equilibrum forces should help The polymer data set is great!

**Weaknesses:**

Probably the biggest weakness is that the authors chose a weak baseline model with TorchMDNet. The entire paper hinges on work from Fu et al. that showed that in their experiments all methods except the NequIP potential were unstable in MD simulation. With many additional tricks and an enormous amount of data they now match NequIP. But will this entire improvement just go away when you apply the ideas to a stronger baseline model like NequIP? I would strongly suggest the authors to repeat the experiments with a better baseline and see if there are still improvements. Otherwise why wouldn't anyone just use NequIP to begin with? The speedup is small and as explained below mostly an artefact of hyperparameters.

Finally, a central weakness it that ET-OREO is pretrained on a massive dataset while the baseline methods that are stable can do this from a much much much smaller dataset.

Another big weakness is that this unexplained improvement from ET-ORE to ET-OREO simultaneously adds the equilibrum and the denoising objective. Why no ablation study to understand which of them is the important one?

The inference speed comparison to NequIP is not thorough. NequIP has shallow pareto curves on accuracy vs efficiency which is well known, i.e. a L=2 models gets almost exactly the same accuracy as L=3, but is much faster (see original NequIP paper). The paper compares to a NequIP model that is solely optimized for accuracy, not for efficienc. This would be helpful to compare to. In addition, NequIP modern versions of NequIP now exists that are faster, e.g. Allegro or MACE.

The presentation in addition has several obvious, outlined item-by-item below:

- "they are locally well-defined for atoms and generalizable across various molecules" --> this statement is clearly not true in this generality. In what way is a force generalizable across molecules? Needs clarification what is meant here?

- following immediately: "they reflect quantum mechanical interactions among atoms and expose the underlying quantum structures of molecules;" --> extremely generic, if even true, atoms should be electrons here...

- "By harnessing these factors, we posit that learning from atomic forces can
57 guide the model to learn a comprehensive and unified landscape of molecular conformations across
58 different molecules, datasets, and equilibrium states" --> this has been explored for decades in every single paper training interatomic potentials, leveraging atomic forces is in now way a novel idea, this needs a strong rephrasing.

- "We introduce a novel force-centric molecular conformation pretraining paradigm that trains
87 a unified conformation representation encompassing both equilibrium and off-equilibrium
88 molecules" this is a vast overstatement, see above

- 'The above-mentioned papers followed the same learning paradigm, where different models are
116 trained for different molecules. Consequently, these methods cannot be easily generalized to unseen
117 molecules" --> there have been many efforts at building general-purpose force-fields, see e.g. the ANI series.

- ". Despite their impressive accuracy in static prediction, their ability
119 of running simulations were not reported until [26] showed that the increased static accuracy does not
120 translate to better simulation performance" again, horrendously false statement, many of the MLIP papers you cite ran various simulations, across diverse and difficult phenoma from ionic diffusion to heterogeneous catalysis to small molecules.

- "In molecular dynamics (MD) simulations, the molecular
149 conformations are moved according to forces and the thermostat of the simulation." what about NVE simulations that don't have a thermostat?

- "For equilibrium molecular conformations x ∼ E , we assume their forces to be zero and impose a
zero-force regularization:" --> what is the reasoning behind this, if you *have* the forces anyway, they will be zero at equilibrium, so you're just again training on forces, again what this community has been doing since the 60s.

- the denoising objective seems like it has obvious failure cases: perturbing a relaxed structure and argueing the force must go into the direction of the GS completely ignores the many-bodyness of the PES, discussing these would be helpful

- a lot of the work seems to be based on improving one or two specific previously introduce pretraining objectives for equilibrium data. In the opening of the manuscript, these are cited several times, but never explained. This makes the manuscript difficult to read.

**Questions:**

N/A

---

> ### Author Response · Authors · 2023-08-19
>
> Dear Reviewer,
>
> We appreciate the time you've taken to review our work. We kindly direct your attention to the points we made in our reply.
>
> - Our contribution is to the pre-training framework, not the model architecture itself. ET-OREO has significantly improved the performance of ET, and as we have shown in the tables we submitted in the reply, outperforms NequIP by as much as 40%, with significantly reduced training time and inference time and greater versatility to finetune on different molecules. This should be enough for one to say that the pre-training framework of ET-OREO is effective.
>
> - Again, our contribution is the pre-trained framework that hinges on forces. This includes the joint training of forces on off-equilibrium data and the de-noising objective on equilibrium data as an approximation. We will make changes to our claims regarding training with forces, see the end of this reply.
>
> - It is also well-known that pre-trained models work better in larger datasets with deeper models and more parameters. Of course, NequIP is shown in the existing paper to be a strong baseline when for a small model data are limited, but neither is the case in the context of pre-training. We re-iterate that our main contribution is the pre-training framework, which in principle is independent of the model architecture of choice, and the choice of ET is based on the consideration of scalability, inference speed, and the size of model parameter space. The inference time of NequIP makes it more difficult for a pre-training task.
>
> - We are still training a NequIP model with our pre-training framework. It takes a significantly longer time, and we will share the results as soon as possible.
>
> In addition to the results we reported previously, upon further tuning ET-OREO to 2000 epochs, we recorded even better performance  metrics on ET-OREO than previously shared:
>
> - Aspirin: 1.0 meV/A (52% improvement over NequIP's 2.1 meV/A)
> - Ethanol: 1.0 meV/A (28% improvement over NequIP's 1.4 meV/A)
> - Naphthalene: 0.9 meV/A (18% improvement over NequIP's 1.1 meV/A)
> - Salicylic Acid: 0.9 meV/A (10% improvement over NequIP's 1.0 meV/A)
>
> Thank you for your constructive feedback, and we are looking forward to your reply.
>
> ----------------------
> Here's the revised version of the third paragraph in the Introduction where the reviewer expressed several concerns about the claim of the paper and language. We have clarified the concept of forces being generalizable and removed the use of language that suggests that we were the first to leverage forces.
>
> In this paper, we incorporate both off-equilibrium and equilibrium conformations into a unified representation learner.
> We propose a new pre-training model, "ET-OREO'" for 3D molecular conformations.
> This model integrates both equilibrium and off-equilibrium data from multiple large-scale datasets.
> Its training objective hinges on \emph{atomic forces}, defined as the negative gradient of a molecule's potential energy with respect to atomic coordinates.
> Atomic forces exhibit several notable properties: (1) they are \emph{physically well-defined} observable, i.e., the force acting on an atom is determined solely and uniquely from its local environment, defined as the real-space distribution of its neighboring atoms; (2) they are generalizable across various molecules in the sense that atoms from different molecules that have the same local environment should experience the same atomic forces; and (3) they can unify equilibrium and off-equilibrium data, as equilibrium data can be conceptualized as local minima in the latent (configuration) space with zero forces, while off-equilibrium data aid the model in more accurately characterizing the high-energy chemical space beyond equilibrium. Among these points, (2), which is the direct consequence of (1), makes atomic forces fundamentally different from potential energy which is defined for the whole system under consideration (molecules) and can only be determined up to an additional constant. Therefore, a predictive model of atomic forces is transferable, i.e., in principle, it can be used directly for molecules of any size (number of atoms). This advantage makes the "learning atomic forces"' fundamentally different from the traditional approach in which a fictitious concept of ``atomic energy'' must be defined, predicted, and combined to obtain the total potential energy of the whole system.

---

> > ### Comment · Reviewer_EQHq · 2023-08-19
> >
> > Thank you for the comments. I have adjusted my score. I believe there remain issues on how useful this is in practice and the weak baseline chosen, as outlined extensively above. This is a lot of pretraining for an improvement of 2.1 meV/A which likely wouldn't even be noticeable under DFT (the errors reported here are already way beyond DFT accuracy, no practitioner would ever care about this difference, let alone be willing to do the pretraining). As such I believe this method will see limited adoption, but may still be a valuable contribution to improve pretraining strategies and objectives.

---

> > > ### Author Response · Authors · 2023-08-20
> > >
> > > Thank you for the comment and score adjustment. We agree that continuing to improve accuracy with respect to DFT has little incremental value as the latter is after all itself a computational approximation, which was why we put much emphasis on the relatively short time required to develop an accurate forcefield and versatility of ET-OREO being a pre-trained model.
> > >
> > > Further, we are trying to explore more applications of our method and better evaluation metrics. On Ethanol and Naphthalene, further finetuned ET-OREO can achieve lower h(r), signifying more similar bond distribution to that of a DFT simulation. We are also evaluating if our model can improve other more interesting applications other than force prediction accuracy. We have some initial positive results, and while we can't share the results in this paper in time, we are excited to include them in future work.
> > >
> > > Thank you.

---

### Author Rebuttal · Authors · 2023-08-10


First, we thank the reviewers for their time and effort to provide valuable feedback. Below, we take the opportunity to discuss your points of concern.

**Our contribution is focused on the pre-training paradigm, instead of the model architecture.**
 NequIP has a great performance with limited training data and outperforms TorchMD-Net. The major novelty of this paper is orthogonal to NequIP – that we proposed a pre-trained framework that should ideally work for arbitrary deep learning models with sufficient expressivity, including NequIP. However, for practical reasons, we chose TorchMD-Net as the base model for efficiency:  we discovered that the training times for TorchMD-Net and NequIP differ significantly (Table R1), preventing us from conducting experiments using NequIP, even for lower basis.

**Also in the paper, we focus on fastly developing an accurate forcefield from the pre-trained ET-OREO with limited training epochs. With the more epochs, ET-OREO outperforms NequIP.**

Rarely discussed in recent deep learning forcefield literature are the epochs required to achieve the reported performance. To achieve the performance reported in our paper, NequIP needs to be trained for maximally 2000 epochs for 10k data, while ET-OREO needs only be finetuned for 300 epochs. In Table R2, we compare NequIP and ET-OREO specifically at different epochs. ET-OREO, when further trained, improves ~30% over NequIP.

The reason that we chose to report only 300 epochs in the paper is because the purpose of developing pre-trained models is that it can serve as an efficient base forcefield that can be efficiently adapted to many molecules. ET-OREO helps achieve NequIP-level performance at 300 epochs and further improved performance at 1000, at an already faster inference time per epoch.
In addition to the inference speed per training step, the number of epochs required for convergence also suggests a higher difference in time required to develop a forcefield from ET-OREO and NequIP. During the rebuttal time, we couldn’t train NequIP fully to 2000 epochs ourselves. However, for a quick reference, NequIP (l=1) takes 21 hours to train for 1000 epochs, (l=2) takes 35 hours, on a single A100 for aspirin.

Of course, NequIP is still a better performing model compared with TorchMD-Net without pre-training, and we could reasonably assume that NequIP pre-trained with the ET-OREO framework could also have more impressive performance. In the future, we will incorporate NequIP and its subsequent more efficient variants into our work.

**Force-centric Pre-training.**

The idea of training on atomic forces is not new; However, our motivation and contribution is using forces as a principle to unify the pre-training of molecules of many different domains.

Previously, deep learning potentials have achieved impressive accuracy on limited data from single molecules. However, as witnessed in recent development in NLP and CV, with the massive availability of data and highly efficient model, we have the potential to push the boundary further. Models such as LLama and ChatGPT are not necessarily the best-performing chatbot model when data is limited. In fact, with limited data, I doubt if they can outperform simple rule-based models, which have been going on for decades. However, combined with efficient training strategies on massive corpora from the internet from multitudes of different domains, they transform the whole NLP landscape as we know it.

Traditionally, people develop models trained on potential energy, or the combo of potential energy and forces. Indeed, ANI-1 has an impressive performance on a large collection of molecules, trained solely on potential energy. However, we find this approach difficult when extending to multiple sources of data. Within MD17 itself, it already presents a challenge (See our Supplementary). One potential reason why training on potential energy could fail might be the software differences when obtaining the DFT predictions; The Supplementary of our paper and of [1] have some theoretical insights as to why this happens from an optimization perspective. Therefore, we resort to training exclusively on forces.

Finally, we incorporate the data where forces are assumed zero, where we can apply the de-noising objectives to further provide useful signals to the model. In our paper, we find a way to unify the pre-training of these data under one force-centric principle in the context of deep learning models.

**We will also make our languages and claims more concise, as the following:**
- Generalizability is in reference to the machine learning model and not the forces being trained on. We mean that one model can make predictions for many different molecules. We will make revisions to make the language more precise.
- Indeed, electrons were the better word to use when speaking from a strictly quantum mechanics standpoint as DFT solves wavefunctions for electrons rather than atoms. This slip-up in language is because the forces calculated later in DFT and used in training here are on the atomistic scale, not the electron scale.
- Most papers, and the ones cited by the reviewer, train one model for one atomistic system (one model for one molecule), and the trained model can thus only work for force predictions for one molecule. In doing this NequIP has done extremely well. What we have done is train one pre-trained model to many different atomistic systems or molecules, with relatively little efforts on fine-tuning. In Section 4.2 we also deploy ET-OREO on large polymers unseen in the training data. We will rephrase to make our claim more precise.
- Perturbed relaxed conformations don’t necessarily go back to the original structure, however it is the best guess we can have without further information. This comes from the Gaussian assumption from which the widely used MSE objective is also derived.

[1] https://www.science.org/doi/10.1126/sciadv.1603015

---

### Decision · Program_Chairs · 2023-09-21

**Decision:**

Accept (poster)

**Comment:**

The authors introduce a pre-training paradigm for molecular representation learning that incorporates both equilibrium and off-equilibrium conformations. As strengths, this paper is well-written and well-organized, effectively presenting the significance of force-centric pretraining for molecular systems. The proposed framework demonstrates comprehensive and impressive results in molecular dynamics simulations of small molecules and polymers, as well as in quantum chemistry property predictions. While other pre-training methods have achieved similar state-of-the-art results, the authors introduce a new framework for including off-equilibrium conformations in the pre-training objective. This work also contributes a new dataset of large polymers Poly24, which reviewers agreed will add value to the research community.